# Identification of QTLs with effects on seed coat appearance in cowpea

Abdoul Moumouni Iro Sodo[1,2]*, Christian Fatokun[1], Bunmi Olasanmi[3], Patrick Obia Ongom[4]*, Ibnou Dieng[1], Ousmane Boukar[4]

1 International Institute of Tropical Agriculture (IITA), Ibadan, Nigeria, 2 Pan African University Life and Earth Sciences Institute (Including Health and Agriculture), University of Ibadan, Ibadan, Nigeria, 3 Department of Crop and Horticultural Sciences, University of Ibadan, Ibadan, Nigeria, 4 International Institute of Tropical Agriculture (IITA), Kano, Nigeria

* abdoulmoumouniirosodo@gmail.com (AMIS); p.ongom@cgiar.org (POO)

## Abstract

Cowpea *(Vigna unguiculata* [L.] Walp.) is a key protein source in sub-Saharan Africa, and seed coat appearance traits such as pigmentation and texture are critical for market segments (MS). We investigated the genetic basis of these traits using 316 recombinant inbred lines derived from a cross between the landrace RP270 (white coat – colorless eye) and the improved variety CB27 (white coat – black seed eye) over two years under regular rainfed conditions. A panel of 2602 high-quality DArTag single nucleotide polymorphisms (SNPs) was used to genotype the RIL population. A total of 30 main QTLs associated with seed coat appearance were identified with phenotypic variation explained ranging from 1.2% to 78.4%. Epistatic analysis revealed 116 significant digenic interactions, highlighting the complex inheritance of pigmentation traits. Based on gene ontology and available literature, we highlighted 12 candidate genes involved in the regulation of seed coat pigmentation in cowpea grains. These findings provide a basis for further research on the genetics of cowpea pigmentation and support future work on map-based cloning of candidate genes and marker-assisted cowpea breeding programs.

## Introduction

Cowpea [*Vigna unguiculata* (L.) Walp.] is a highly valued food crop that contributes numerous nutrients, including protein, carbohydrates, minerals, vitamins, as well as bioactive substances such as isoflavones, sterols, and phospholipids to enrich consumers' diets [1]. Compared to many other food legumes, cowpea proteins are rich in tryptophan and lysine. Consequently, cowpea constitutes part of the dietary protein, especially for people living in rural and semi-urban areas of sub-Saharan Africa (SSA). In Africa, because of the protein in grains, cowpea is referred to as "poor man's meat" [1–3]. The plants thrive in low-fertility soils due to their nitrogen-fixing ability. It is drought-tolerant as it grows well in drought-prone areas, making it

**Data availability statement:** The original contributions presented in the study are included in the paper/Supporting information files.

**Funding:** AVISA project, i.e. Accelerated varietal improvement and seed delivery of legumes and dryland cereals in Africa (AVISA) - AVISA-Transition project. The investment ID is INV-049752.

**Competing interests:** The authors have declared that no competing interests exist.

particularly popular in semi-arid regions of the tropics where many other crops do not perform as well. Even in poor soils with organic matter less than 0.2%, a pH range of 4.5-9.0, and a sand content of up to 85%, cowpea has a remarkable ability to perform relatively better than many other crops [4,5]. As the global demand for cowpea continues to rise, research efforts focusing on developing varieties with higher yield, seed systems, and good agronomic practices are gaining increased attention. As with many other grain crops, the acceptance of cowpea by consumers is mainly driven by the visual appearance of the grains. Cowpea seed coat appearance is important as it determines its marketability. It is crucial to know that creating new promising varieties with high yield and large seed size but not acceptable to end-users is ultimately a waste of time, resources, and efforts for breeders. To make research efforts count, breeders now strategically adopt participatory crop improvement methods where inputs of all stakeholders along the value chain are considered. At such stakeholders' gatherings, the target product profiles are well defined, and breeders select parents to cross based on these. Cowpea MS have been defined based on a combination of end usage, seed coat color, maturity and production environments. The key cowpea end uses that constitute the MS are i) boiled grain, ii) home-made flour, iii) industrial flour, iv) dual-purpose (grain and fodder), v) dual purpose (grain and leaves) and vi) fresh pods and grain. Importantly, seed coat color and texture are among the top essential traits in the target product profiles (TPP) of each defined cowpea MS. Previous studies have reported that consumers make decisions on the acceptability, quality, and presumed taste of the grains depending on appearance [6–8].

Seed coat texture is an important trait that influences the acceptance of cowpea varieties across various regions and communities. In western and central Africa, rough seed coat is preferred because it allows for its easy removal which is an essential step for indigenous food preparations, while in eastern and southern Africa, as well as in some areas of South America, smooth seed coat is preferred since cowpea is often consumed as boiled beans without the need to remove the seed coat [9]. Moreover, in many markets, large seed size is preferred, which leads to price premiums for large, seeded varieties. Among the world cowpea germplasm collection, there is an array of diversity in grain size, coat and eye color, and their distribution patterns.

There are regional preferences for cowpea grain types across Africa. For example, in West Africa, the two most popular grain color types with consumers are white or brown, medium to large seed sizes, while in East and Southern Africa, relatively smaller seeds and brown to red predominate in markets [10].

Seed coat pigmentation is primarily due to the accumulation of metabolites, including flavonoids and anthocyanins biosynthesized via the phenyl-propanoid pathway in the epidermal layer of the coat [11]. These metabolic compounds have been the subject of research due to their exceptional antioxidant properties, which offer potential human health advantages. Furthermore, they are known to affect the flavor profiles of the seeds, making them attractive for certain culinary applications [12].

With the progress in molecular biology techniques, breeders and geneticists have been investigating QTLs associated with important cowpea traits. Few studies have

been carried out to identify the genomic regions associated with seed coat color and texture in cowpea. Investigating cowpea seed coat related traits carries not only significant theoretical importance but also practical relevance for its cultivation and use [8]. In $F_2$ populations, four factors, namely Color Factor (C), Watson (W), Holstein-1 (H-1), and Holstein-2 (H-2) that regulate seed coat color pattern and pod tip color were reported by Harland et al. [13] and Spillman [14]. Xu et al. [15] conducted a study to map flower and seed coat colors in $F_{7:8}$ RILs and identified one locus for each trait. The two loci are tightly linked to each other with a genetic distance of 0.4 cM. Herniter et al. [8] reported seed coat color pattern related loci on three cowpea chromosomes namely, LG7 (C locus), LG9 (H locus), and LG10 (W locus) and identified a total of 35 SNP markers related to seed coat patterns. Prior studies in soybean (*Glycine max)* and common bean (*Phaseolus vulgaris*) noted that seed coat colors and patterns are governed by different Mendelian loci with complex epistatic effects and seed coat pattern phenotypes are controlled by C and T loci. The C locus is the primary locus controlling seed coat color pattern. Pigmentation may not be visible, restricted to the eye, or distributed across the entire seed coat, typically showing a contrast of darker color patterns and lighter color [16–18]. The objective of this study was to identify quantitative trait loci (QTLs) associated with cowpea seed coat appearance traits in a set of recombinant inbred lines (RILs) using a medium-density genetic linkage map. This is the first report to map QTLs for seed coat texture in cowpea, and it identifies novel loci for pigmentation, alongside extensive epistatic interactions.

## Materials and methods

### Genetic materials, glasshouse and field trials

A recombinant inbred line (RIL) population made up of 316 individuals, derived from a cross between RP270 and CB27 was used in the present study. The two parental lines differed in seed size parameters, seed eye color, seed coat texture, and number of days to flower. The line RP270, a landrace from Togo, has white coat color, smooth coat texture, no eye color, with a relatively small sized seed (Approx. 15.5g/100). The second parent CB27, an improved USA variety is characterized by white seed coat, black eyes, rough coat texture, and medium size (Approx. 18.7g/100). The single-seed descent method was employed to advance the population to $F_{6:7}$ RIL in the glasshouse at the International Institute of Tropical Agriculture (IITA), Ibadan, Nigeria. Before each sowing, seeds were treated with hexachlorobenzene (Granox N-M Fungicide Seed Treatment Patch) at 10 g/kg based on the manufacturer's recommendation. The RIL population and the parental lines were planted in the field and glasshouse in trials designed in incomplete block design. There were three replications in the field trials at the research field of IITA (7.50250° N, 3.89411° E). The experimental plots consisted of three rows of 2.0 m in length, with a spacing of 1 m between rows and 0.25 m within rows. The seeds were sown on 13th October 2022 and 4th September 2023 following ploughing and harrowing of the plots. In the glasshouse seeds were sown in plastic pots each containing 5.0 kg sterilized topsoil. A compound fertilizer (NPK 15:15:15) was applied at a rate of 6 g per stand two weeks after planting in the field. The plots were kept insect-free by spraying chlorpyriphos (Thermex 48EC) and lambda-cyhalothrin (Karate 5EC) at 2.0 l/ha and 1.5 l/ha, respectively, at the vegetative, flowering, pod formation, and pod filling stages. Manual weeding was carried out as necessary to ensure no adverse weed interference.

### Seed coat phenotyping and statistical analysis

For both 2022 and 2023 experiments, pods were harvested, threshed and the seeds dried to a constant moisture level in the glasshouse. The seeds that were malformed, damaged physically, diseased, or bruchid infested were eliminated. The seed coat appearance traits, including seed eye color, seed coat color, and seed coat color pattern using visual ratings and later encoded into numeric values. Seed coat texture (smooth, rough) was scored using a headband magnifying glass to ensure precision and consistency in visual examination of the seed coat characteristics. Three seed colors, including white, brown, and black, and five seed coat color patterns were observed. For the seed coat color pattern, examination and classification of the seed were done following the description given by Herniter et al. [8]. The observed seed coat pattern classes were No-eye, Eye 2, Holstein, Watson, and Full-coat (Fig 1 and Table 1). These are

described as follows: no eye (no pigmentation present on the seed coat), Eye 2 (exhibits pigmentation (black or brown) limited to immediately around the hilum), Holstein (characterized by pigmentation (black or brown) with same color randomly dispersed across the seed coat), Watson (characterized by pigmentation (black or brown) around the hilum with lighter color of same pigment covering remaining parts of seed coat) and Full-coat (pigment completely covering the seed coat). Three gene systems, C (Color Factor), W (Watson), and H (Holstein), with simple dominance and epistatic interactions, were used to explain the segregation ratios among the RILs [8,14]. For QTL analysis, the data were

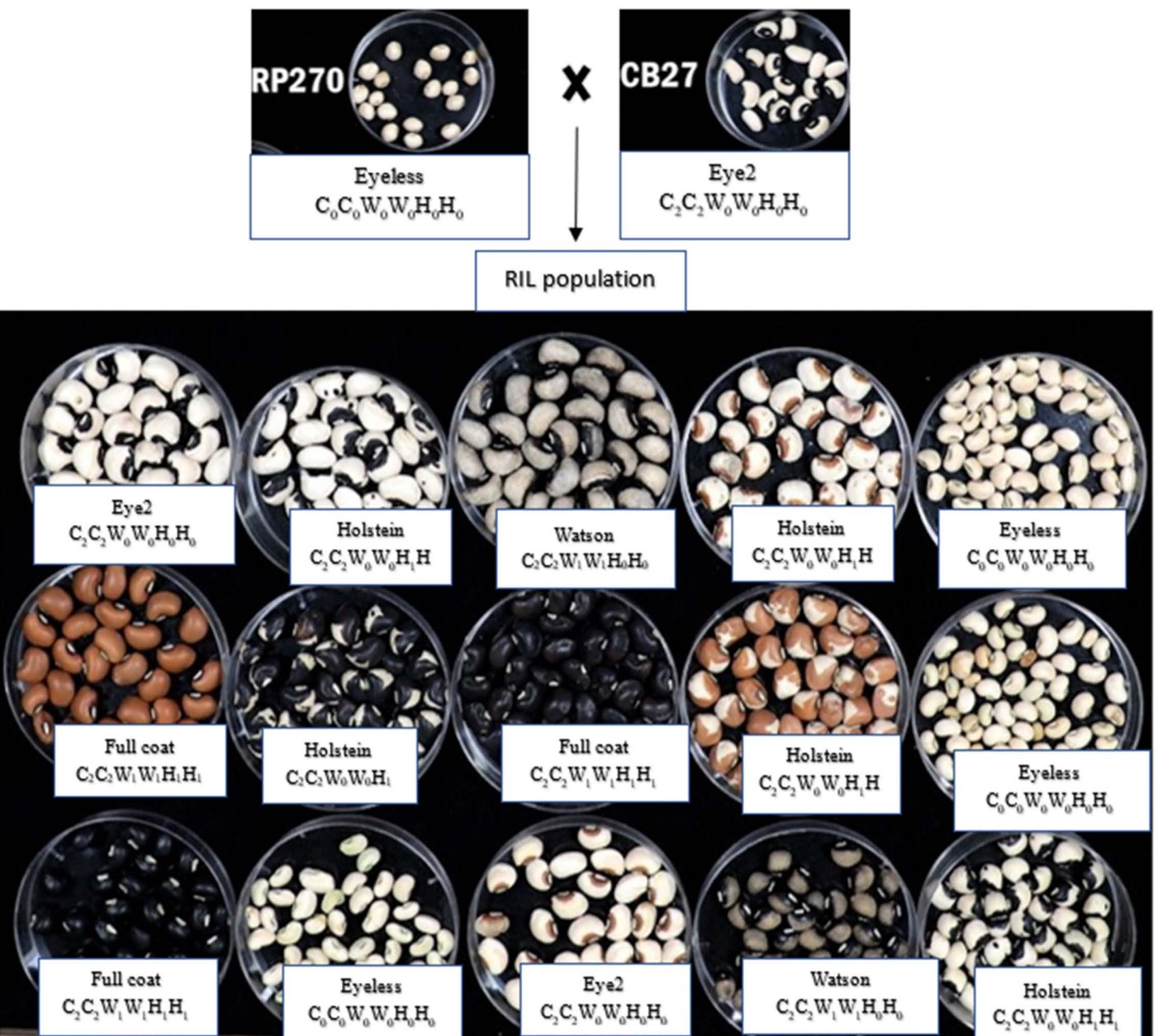

**Fig 1. Seed coat color and distribution patterns in the RIL population.**

Table 1. Phenotypes and expected genotypes, segregation ratios (Seg.ratio), chi-square ($\chi^2$), and probability (Prob).

| | | Phenotypes and expected genotypes | | | | |
|---|---|---|---|---|---|---|
| Parents/RILs | | Phenotype | Proposed genotype | | | |
| RP270 | | No Eye | $C_0C_0W_0W_0H_0H_0$ | | | |
| CB27 | | Eye 2 | $C_2C_2W_0W_0H_0H_0$ | | | |
| F6:7 RILs | | No Eye | $C_0C_0W_0W_0H_0H_0$ | | | |
| | | Eye 2 | $C_2C_2W_0W_0H_0H_0$ | | | |
| | | Holstein | $C_2C_2W_0W_0H_1H_1$ | | | |
| | | Watson | $C_2C_2W_1W_1H_0H_0$ | | | |
| | | Full coat | $C_2C_2W_1W_1H_1H_1$ | | | |
| | | Phenotypes | | | | |
| | | Seed coat texture | | | | |
| Smooth | Rough | Seg.ratio | $\chi^2$ | Pro | | |
| 170 (53.8%) | 146 (46.2%) | 1:1 | 1.82 | 0.177 | | |
| | | Seed eye color | | | | |
| No color | Brown | Black | Seg.ratio | $\chi^2$ | Prob | |
| 100 (31.65%) | 99 (31.31%) | 117 (37.02%) | 1:1:1 | 1.94 | 0.38 | |
| | | Seed coat color | | | | |
| White | Brown | Black | Seg.ratio | $\chi^2$ | Prob | |
| 151 (47.78%) | 77(24.37%) | 88 (27.85%) | 2:1:1 | 1.39 | 0.5 | |
| | | Seed coat pattern | | | | |
| No Eye | Eye 2 | Holstein | Watson | Full coat | Seg.ratio | $\chi^2$ | Prob |
| 100 (31.65%) | 52 (16.14%) | 51(16.14%) | 53 (16.8%) | 60 (18.99%) | 2:1:1: 1:1 | 1.35 | 0.85 |

recorded as follows: each observed pattern was recorded independently, and a score of *1* was given if the trait was present or "0" if the trait was absent. For instance, a line that displays the Full-coat pattern will be scored *1* for that attribute and "0" for all the remaining traits (S1 Table). The segregation ratios were computed based on the expected ratio in the biparental RILs of a 1:1 for traits controlled by single genes. The goodness of fit between observed and expected ratios was evaluated using chi-square ($\chi^2$) tests, with a significance level set at $\alpha = 0.05$.

## DNA extraction, genotyping, and linkage map construction

Leaf tissue samples were collected following the procedure previously described [19,20]. The samples were sent to Intertek Laboratory Sweden, where DNA extraction and subsequent genotyping were conducted, following DArTag technology, a targeted genotyping method that has the capacity to genotype samples with specific or selected sets of SNP markers (https://www.diversityarrays.com/services/targeted-genotying [19]. Data filtering protocols, and linkage map construction were consistent with those applied in our earlier study [20]; detailed methodologies are described therein.

## QTL analysis and candidate gene identifications

Following linkage map construction, QTL analysis was performed using the inclusive composite interval mapping (ICIM) method in QTL IciMapping v4.2 software as previously described [20]. Main-effect QTLs were identified using the inclusive composite interval mapping (ICIM-ADD) function. During the QTL mapping process, some critical parameters were set. These included 1000 shuffles for estimating the critical values of LOD at p = 0.05 of the probability value (P-value) for the permutation option to declare the presence of a significant main-effect QTL. Similarly, to map epistatic QTLs (E-QTLs), the ICIM-EPI option was used with identical step size (1 cM) and PIN (0.0001) parameters. The permutation option to declare

the presence of epistatic-QTL (E-QTL) was set, including 1000 shuffles for estimating the critical values of LOD at p = 0.05 of the probability (P-value). The QTLs explaining 10% or more of the phenotypic variation (PVE ≥ 10%) were considered major, while those explaining less than 10% were classified as minor as previously described [20]. Results from QTL analyses, genes associated with seed coat color and patterns of distribution were identified using flanking markers' positions and cowpea reference genome (IT97K-499–35)- https://phytozome-next.jgi.doe.gov/info/Vunguiculata_v1_1 Vigna unguiculata v1.1 [21], which has a black Eye 1 (C1C1). Based on gene ontology and available literature, the possible putative genes were selected based on their participation in the flavonoid biosynthesis pathway.

### Relative efficiency of marker-assisted selection (RE-MAS)

We estimated additive genetic variance and heritability for binary seed-coat traits using a genomic animal model on the liability (probit) scale. A realized genomic relationship matrix was computed from SNP dosages and used as the additive genetic covariance among RILs [22]. For each trait we fit a univariate threshold animal model in a Bayesian framework with MCMCglmm, fixing the residual variance to 1 on the latent scale [23]. Posterior mean additive variance on the latent (probit) scale was converted to the observed binary scale using the QGglmm framework, which implements the integrals required for probit GLMMs [24]. For each trait the variance explained by candidate QTL (VM) was computed on the observed scale. VM was computed using LD-aware quadratic form (joint approach) to correctly account for inter-marker covariance and avoids double-counting of variance contributed by correlated loci. To compare selection schemes, we computed three accuracies:

- Phenotypic accuracy (PS): $r_{P,A} = \sqrt{h^2 obs}$
- Marker-only accuracy (MBS): $r_{M,A} = \sqrt{VM/VA}$
- Index accuracy for combined marker + phenotype (MAS) —approximate index accuracy: $r_I = \sqrt{(VA + VM)/VA}$.

From these we computed:

$RE_{MBS:PS} = \frac{r_{M,A}}{r_{P,A}}$, $RE_{MAS:PS} = \frac{r_I}{r_{P,A}}$ following selection-index theory [25] and MAS efficiency theory [26]. All analyses were performed in R (packages: AGHmatrix/Gmatrix, MCMCglmm, QGglmm).

## Results

### Phenotypic variation

The seed coat texture segregated 170 smooth:146 rough in the ratio of 1:1 as expected in the biparental RIL population with chi-square value 1.82 less than the tabulated value (3.84) at P‹0.05. This suggests that seed coat texture is controlled by a single gene. However, for the seed eye color and coat pigmentation, the RIL population exhibited a diverse array of phenotypes. Black eye color was the most prevalent (37.02%), followed by no eye-color (31.65%), and brown color (31.31%). The RIL population segregated in the ratio of 1:1:1 and a chi-square 1.94 with a p-value of 0.38 (Table 1), indicating that there is likely a single region that controls seed eye color in cowpea. White seed coat color was most common among the RILs, accounting for 47.78%, followed by black (27.85%), and brown colors (24.37%), exhibiting a segregation ratio of 2:1:1 with a chi-square value of 1.39 and a p-value of 0.5. For seed coat color pattern, five categories of seed coat patterns were observed among the RILs in the ratio 2:1:1:1:1 for no eye, Eye 2, Holstein, Watson, and full coat, respectively, with a chi-square value of 1.35 and a p-value of 0.85.

### Linkage map

One thousand and eighty-three (1083) high-quality SNP markers with confirmed positions in the cowpea genome were used for constructing a genetic linkage map that covered a total of 794.7 cM, with a cumulative average distance of 0.74 cM between adjacent markers. The number of SNP markers mapped on each of the eleven *V. unguiculata* linkage groups

ranged from 65 for VuLG6–150 for VuLG3, spanning from 53.0 cM for VuLG11 to 120.1 cM for VuLG3 with an average of 72.25 cM. A variation in marker density was observed among the linkage groups. The highest density of 3.47 cM/Mb was found on VuLG4 and VuLG6 followed by VuLG11 (3.40 cM/Mb), while VuLG1 displayed the lowest marker density of 2.66 cM/Mb (S2 Table).

## QTL mapping

A total of 30 QTLs with effects on seed eye color, seed coat color patterns, and seed coat texture were distributed on five chromosomes including Chr5, Chr7, Chr8, Chr9, and Chr10 explaining phenotypic variations of 1.2%–78.4%, with LOD scores ranging from 3.0 to 2583.7 (Table 2 and S1 Fig). The Manhattan plots (Fig 2) provide visual representations of these findings. Twenty-four QTLs were detected for pigmentation of seed eyes, seed coat, as well as their distribution patterns on the latter. Among these, three QTLs (qNo_eye-5–1, qNo_eye-7–1, and qNo_eye-7–2) were linked to the colorless seed eye, identified on Chr5 and Chr7, explaining phenotypic variances ranging from 1.2% to 78.4%, with LOD scores between 6.9 and 289.0. Two major QTLs (qblk_eye-5–1 and qblk_eye-7–1), were associated with black seed eye color identified on Chr5 and Chr7, respectively. These QTLs explained 42.8% and 24.05% of phenotypic variance, with LOD scores of 56.3 and 35.7, respectively. Additionally, one significant QTL for brown seed eye color (qbr_eye-7–1), which explained 17.3% of the phenotypic variance and showed LOD score of 28.7 was mapped to Chr7. Three QTLs were associated with black seed coat. Two of these were major QTLs on Chr5 and Chr7 and one minor QTL on Chr9, with PVE values ranging from 3.5% to 26.3% and LOD scores of between 4.3 and 27.5. One major and one minor QTLs on Chr7 and Chr9 were associated with brown seed coat color, accounting for 13.7% and 3.0% of phenotypic variance, with LOD scores of 16.1 and 4.2, respectively. One unique major QTL was identified for white seed coat color (qwht_c-7–1) on Chr7, explaining 77.3% of the phenotypic variance with an exceptionally high LOD score of 2583.7, suggesting a strong and significant association between this QTL and white seed coat color.

Four QTLs were mapped for Eye 2 pattern on Chr7 and Chr10, explaining phenotypic variance of between 1.9% and 21.5%, with LOD scores ranging from 3.0 to 27.9. For the Holstein seed coat pattern, three QTLs were detected on Chr7, Chr9 and Chr10, with PVE values ranging from 8.1% to 21.3% and LOD scores between 9.2 and 24.4 while two QTLs were identified for the Watson pattern on Chr7 and Chr10 explaining 9.2% and 19.0% of the phenotypic variance, with LOD scores of 12.8 and 24.3, respectively. Regarding the full coat color patterns, three QTLs were detected on Chr7, Chr9, and Chr10, with PVE values ranging from 9.5% to 19.2% and LOD scores between 10.5 and 20.3. Multiple traits namely eye 2 pattern, Holstein pattern, Watson pattern, Full coat, seed black eye, seed brown eye, black seed color, brown seed, and white seed were co-located on Chr7 over a distance of 40-45cM and lying between markers 2_20060 and 2_28580 (Fig 3).

Three QTLs were identified for seed coat texture on Chr8 and Chr10, explaining phenotypic variance between 3.1% and 26.9%, with LOD scores ranging from 3.5 and 25.6 respectively (Table 2, Fig 3). The significant QTLs presented in Table 2 were consistently identified in both years. The detailed results from the single-year analyses are provided in S3 Table.

## Epistatic QTLs

A total of 116 pairs of digenic interactions were mapped across all eleven cowpea chromosomes for seed coat pigmentation patterns (eye color, coat color, and color distribution patterns), with LOD values ranging from 3.0 to 2331.7 (Fig 4 and S4 Table). These epistatic QTLs showed an average PVE of 6.92% and ranging from 0.19%to 62.04%, confirming their smaller contribution compared to main-effect QTLs. The highest number of epistatic QTL pairs (52) was observed for the no eye color pattern and white seed coat color. Several traits showed significant epistatic interactions on chromosomes 5, 7, 9, and 10, suggesting possible pleiotropic effects. Notable interactions include: Black seed eye: Chr5-Chr7 interaction explaining 62.0% of phenotypic variance (PVE) with the highest LOD value of 2331.7; Brown seed eye: Chr5-Chr7

**Table 2. List of putative QTLs for seed coat pigmentations and seed coat textures.** QTL designation followed by a symbol of the trait name and number of the chromosome the QTL is located, Chr: chromosome, Pos: position of the QTL, L-marker (left marker), R-marker (right marker), phenotypic variance (PVE) explained by the QTL, additive effect (Add).

| Trait | QTL | Chr | Pos (cM) | L-Marker | R-Marker | LOD | PVE (%) | Add |
|---|---|---|---|---|---|---|---|---|
| No eye | qNo_eye-5–1 | 5 | 62 | 2_28104 | 2_16977 | 71.2 | 8.5 | 0.4 |
| | qNo_eye-7–1 | 7 | 39 | 2_43619 | 2_20060 | 289 | 78.4 | 0.5 |
| | qNo_eye-7–2 | 7 | 48 | 2_28580 | 2_17387 | 6.9 | 1.2 | 0.1 |
| Eye 2 | qEy2-7-1 | 7 | 41 | 2_20060 | 2_28580 | 8 | 5.9 | −0.1 |
| | qEy2-8-1 | 8 | 28 | 2_27575 | 2_03284 | 3 | 1.9 | 0.1 |
| | qEy2-10-1 | 10 | 15 | 2_07481 | 2_05168 | 27.9 | 21.5 | −0.2 |
| | qEy2-10-2 | 10 | 34 | 2_52788 | 2_09243 | 5.7 | 3.7 | 0.1 |
| Holstein | qHol-7–1 | 7 | 45 | 2_20060 | 2_28580 | 9.2 | 8.1 | −0.1 |
| | qHol-9–1 | 9 | 42 | 2_10786 | 2_09349 | 24.4 | 21.3 | −0.2 |
| | qHol-10–1 | 10 | 15 | 2_07481 | 2_05168 | 20.7 | 17.6 | 0.2 |
| Watson | qWat-7–1 | 7 | 40 | 2_20060 | 2_28580 | 12.8 | 9.2 | −0.1 |
| | qWat-10–1 | 10 | 16 | 2_05168 | 2_00712 | 24.3 | 19 | −0.2 |
| Full coat | qfull_C-7–1 | 7 | 40 | 2_20060 | 2_28580 | 10.5 | 9.5 | −0.1 |
| | qfull_C-9–1 | 9 | 44 | 2_02101 | 2_42066 | 17.3 | 16.4 | 0.2 |
| | qfull_C-10–1 | 10 | 15 | 2_07481 | 2_05168 | 20.3 | 19.2 | 0.2 |
| Seed black eye | qblk_eye-5–1 | 5 | 13 | 2_34048 | 2_04598 | 56.3 | 42.8 | −0.3 |
| | qblk_eye-7–1 | 7 | 41 | 2_20060 | 2_28580 | 35.7 | 24 | −0.3 |
| Seed brown eye | qbr-eye-7–1 | 7 | 41 | 2_20060 | 2_28580 | 28.7 | 17.3 | −0.2 |
| Black seed | qblk_S-5–1 | 5 | 13 | 2_34048 | 2_04598 | 27.5 | 26.3 | −0.2 |
| | qblk_S-7–1 | 7 | 41 | 2_20060 | 2_28580 | 17.6 | 16.8 | −0.2 |
| | qblk_S-9–1 | 9 | 29 | 2_04052 | 2_52520 | 4.3 | 3.5 | 0.1 |
| Brown seed | qbr_S-7–1 | 7 | 41 | 2_20060 | 2_28580 | 16.1 | 13.7 | −0.2 |
| | qbr_S-9–1 | 9 | 43 | 2_14820 | 2_02101 | 4.2 | 3 | 0.1 |
| White seed | qwht_S-7–1 | 7 | 42 | 2_20060 | 2_28580 | 2583.7 | 77.3 | 0.5 |
| Seed coat texture | qSmh-8–1 | 8 | 18 | 2_24183 | 2_10844 | 21.9 | 21.4 | 0.2 |
| | qSmh-8–2 | 8 | 75 | 2_24777 | 2_02595 | 3.7 | 3.1 | −0.1 |
| | qSmh-10–1 | 10 | 14 | 2_16985 | 2_07481 | 24.8 | 25.3 | 0.2 |
| | qRgh-8–1 | 8 | 18 | 2_24183 | 2_10844 | 20.7 | 20.4 | −0.2 |
| | qRgh-8–2 | 8 | 75 | 2_24777 | 2_02595 | 3.5 | 3.0 | 0.1 |
| | qRgh-10–1 | 10 | 14 | 2_16985 | 2_07481 | 25.6 | 26.9 | −0.2 |

interaction accounting for 55.1% PVE (LOD 103.9); Brown seed: Chr5-Chr7 interaction explaining 49.8% PVE (LOD 77.4); Holstein pattern: Chr7-Chr10 interaction with 41.6% PVE (LOD 26.6); Watson pattern: Chr9-Chr10 interaction explaining 36.9% PVE (LOD 22.3); Black seed: Chr5-Chr7 interaction with 34.7% PVE (LOD 20.4); Full coat pattern: Chr7-Chr10 interaction explaining 29.9% PVE (LOD 13.1); and Eye 2 pattern: Chr7-Chr10 interaction with 29.2% PVE (LOD 13.7).

### Identification of candidate genes

In this study, all the model genes were retrieved from the phytozome v1.1 database (http://phytozome.jgi.doe.gov/) and the interPro portal was used for gene models along with their functional annotations. These genes (Table 3 and S5 Table) were identified on each locus by considering the level of relative expression of the flavonoid biosynthesis pathway. No eye color was found in overlapping regions on Chr5 and Chr7, where the genes *Vigun05g236200, Vigun05g237400,* and *Vigun07g057300* coding for MYB transcription factors and the WD40-repeat-containing domain (Chr5) were identified

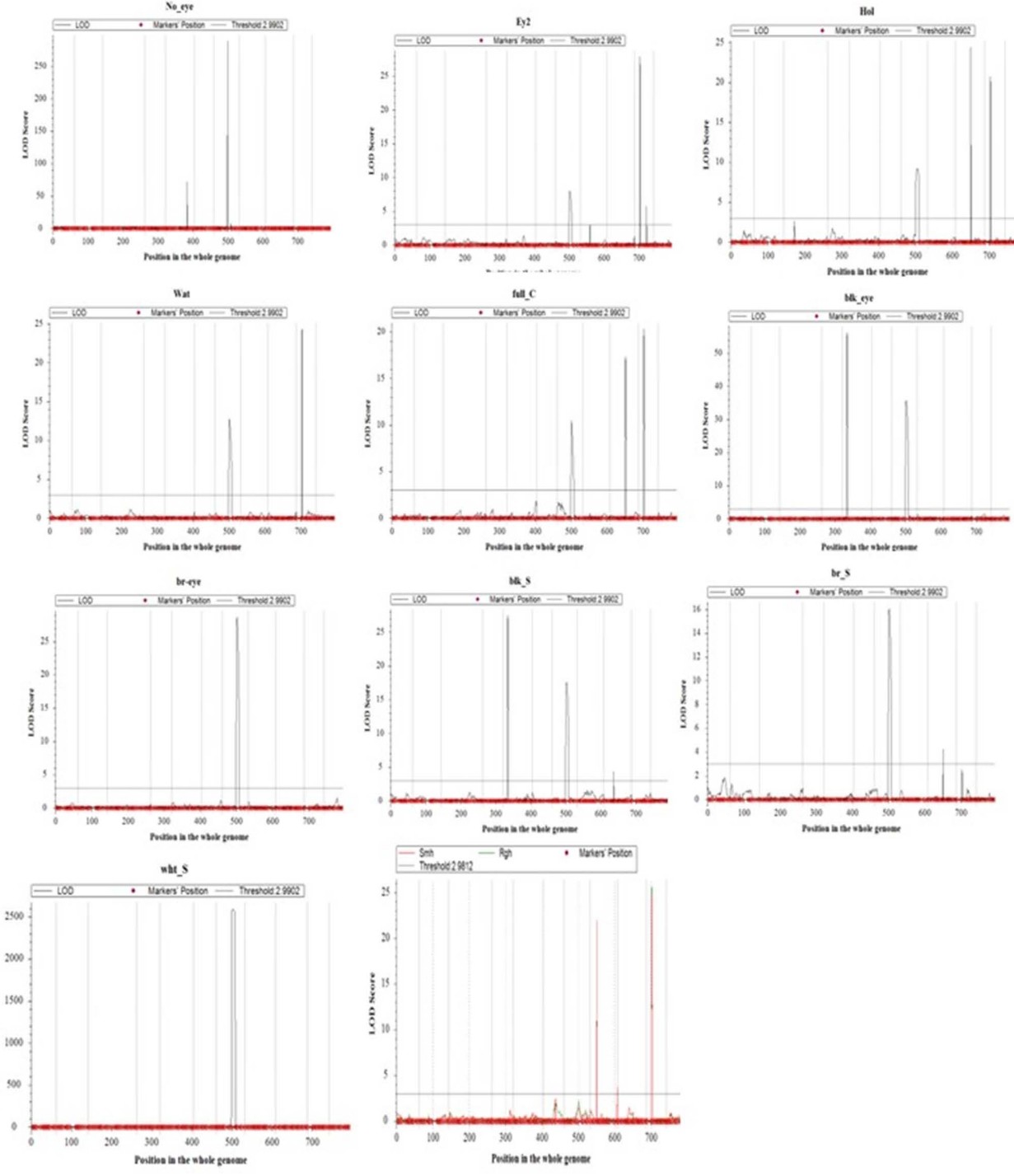

**Fig 2. Manhattan plots of QTL mapping results.** Each plot shows the LOD scores of markers for the respective traits. Significant QTLs are indicated by peaks that exceed the LOD score threshold, marked by a horizontal black dashed line. Hol: Holstein, Wat: Watson; Full-C-7: Full coat, Blk-eye: black eye, Br-eye: Brown eye, br-S: Brown seed coat, wht-S: White seed coat, Smh: Smooth, Rgh: rough.

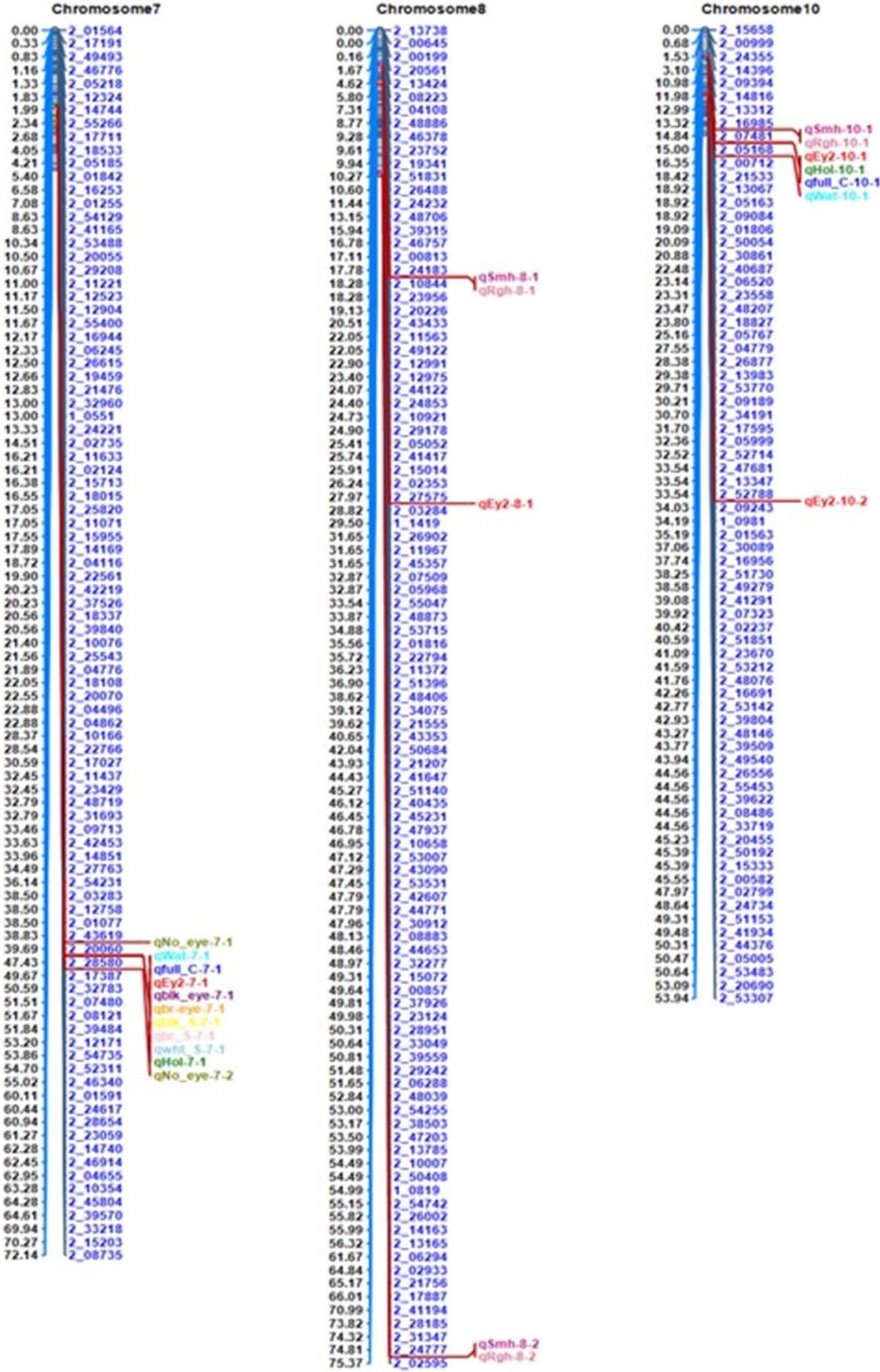

**Fig 3. Cluster of QTLs associated with seed coat appearance.** The QTLs for traits are represented using their QTL names. Hol: Holstein, Wat: Watson, Full-C-7: Full coat, Blk-eye: black eye, Br-eye: Brown eye, br-S: Brown seed, wht-S: White seed, Smh: Smooth, Rgh: rough.

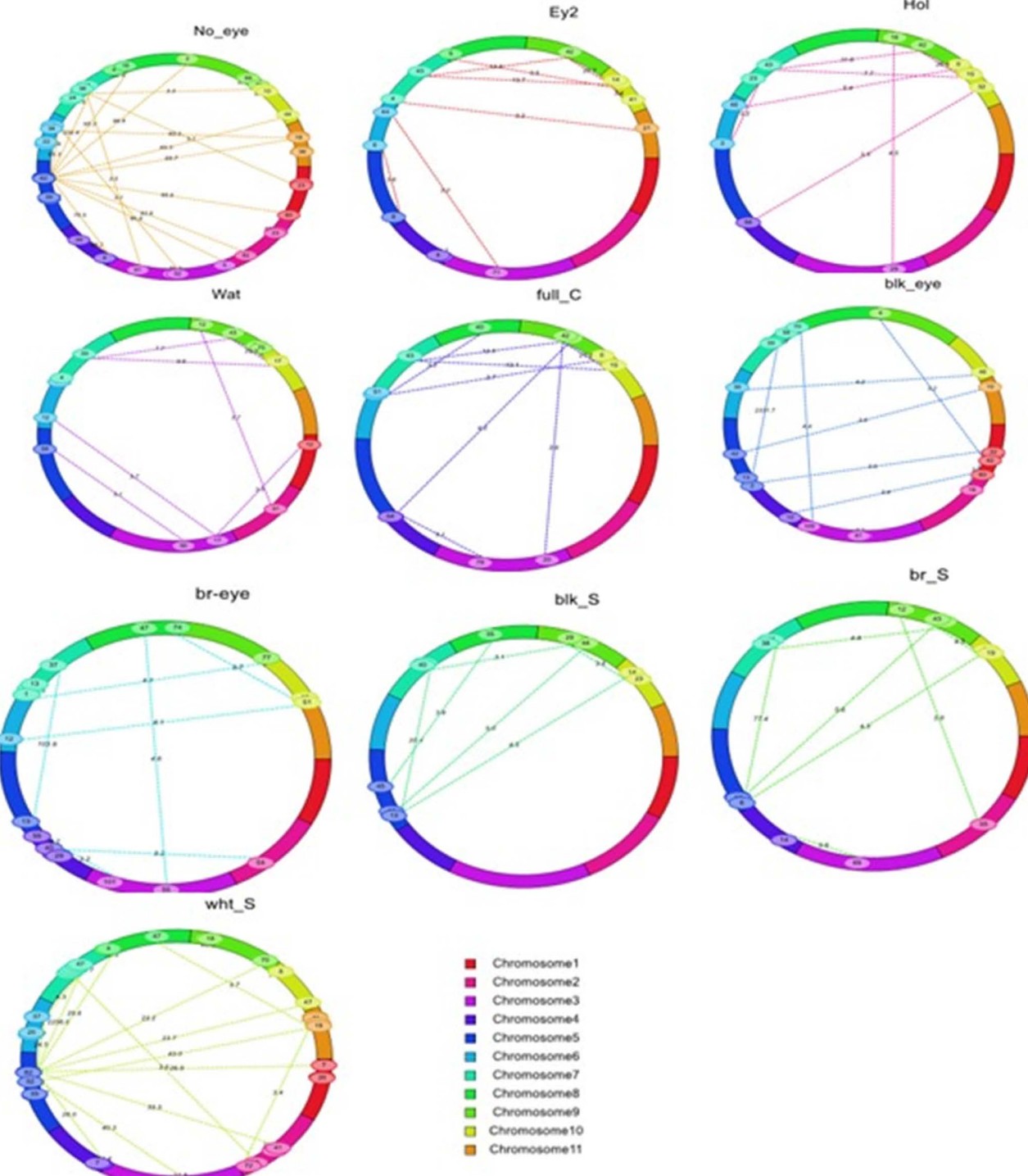

**Fig 4. Cyclic demonstration of epistatic interactions of QTLs linked to cowpea seed eye color, seed coat color and seed coat color patterns.** The cowpea's 11 chromosomes were represented by 11 colors in the ring, with the numbers in the ovals indicating the positions of markers. The dotted lines imply the interacting marker pairs on different or on the same chromosomes due to epistatic interaction. The LOD scores of the epistatic QTLs are indicated by the dotted lines.

**Table 3. Candidate genes identified for seed coat pigmentation traits in cowpea. Chr: chromosome, Ref: reference.**

| Traits | Chr | Gene ID | Putative Gene | Gene Functional Annotation | Ref. |
|---|---|---|---|---|---|
| No eye | 5 | Vigun05g236200 | MYB transcription factors | DNA-binding transcription factor involved in the regulation of secondary metabolite biosynthesis, particularly flavonoids and anthocyanins. Functions in seed coat color determination | [27] |
| | | Vigun05g237400 | WD40-repeat-containing domain | Protein binding; oxidoreductase activity; peroxidase activity; response to hydrogen peroxide; regulation of secondary metabolism and transcription. Associated with flavonoid biosynthesis and seed coat color in legumes. | [8] |
| | 7 | Vigun07g057300 | MYB transcription factors | DNA-binding transcription factor involved in the regulation of secondary metabolite biosynthesis, particularly flavonoids and anthocyanins. Functions in seed coat color determination | [27] |
| Eye2, Holstein, Watson, Full coat, Black eye, Brown eye, Black seed, Brown seed, White | | Vigun07g068100.1 | AP2/ERF domain-containing protein | DNA-binding transcription factor involved in ethylene signaling and may influence pigment biosynthesis through transcriptional regulation | [28] |
| | | Vigun07g072800.1 | WD40/YVTN repeat-like-containing domain | Protein binding; oxidoreductase activity; peroxidase activity; response to hydrogen peroxide; regulation of secondary metabolism and transcription. Associated with flavonoid biosynthesis and seed coat color in legumes. | [8] |
| | | Vigun07g103700.1 | WRKY domain | DNA binding, Regulation of transcription | [28] |
| Eye2, Holstein, Watson, Full coat | | Vigun10g165100.1 | AP2/ERF domain | DNA-binding transcription factor involved in ethylene signaling and may influence pigment biosynthesis through transcriptional regulation | [28] |
| | 10 | Vigun10g165200.1 | Ubiquitin-related domain | Involved in tagging proteins for degradation via the proteasome. Regulates gene expression and signal transduction by controlling the stability of transcription factors. | [8] |
| Eye2 | 10 | Vigun10g134800.1 | WRKY domain | DNA binding, Regulation of transcription | [28] |
| Holstein | 9 | Vigun10g165100.1 | AP2/ERF domain | DNA-binding transcription factor involved in ethylene signaling and may influence pigment biosynthesis through transcriptional regulation | [28] |
| | | Vigun10g165200.1 | Ubiquitin-related domain | Ubiquitin protein ligase activity (GO:0061630), Ubiquitin-dependent protein catabolic process (GO:0006511), Proteasome complex (GO:0000502) | [8] |
| Black seed | 9 | Vigun09g079900.1 | MYB transcription factors | DNA-binding transcription factor involved in the regulation of secondary metabolite biosynthesis, particularly flavonoids and anthocyanins. Functions in seed coat color determination | [27] |

as putative genes. Eye 2, Holstein, Watson, full coat, black eye, brown eye, black seed, brown seed, and white seed were found in the same overlapping region on Chr7 defined by the same markers (2_20060 and 2_28580), where the genes *Vigun07g068100, Vigun07g072800,* and *Vigun07g103700* coding for AP2/ERF domain-containing protein, WD40/YVTN repeat-like-containing domain, and WRKY domain, respectively, were identified as putative genes. On Chr10, Eye 2, Holstein, Watson, and full coat were identified in a similar region by the flanking markers (2_07481 and 2_05168), where the genes *Vigun10g165100* and *Vigun10g165200*, coding for the AP2/ERF domain and Ubiquitin-related domain, were found as putative genes. Eye 2 was detected on Chr10 defined by the markers (2_52788 and 2_09243), where gene *Vigun10g134800* coding for the WRKY domain was found as a putative gene and Holstein was detected in an overlapping region on Chr10 where the genes (*Vigun10g165100* and *Vigun10g165200*) coding for AP2/ERF domain and

Ubiquitin-related domain were found to be candidate genes. The black seed coat was mapped on Chr9, where the gene *Vigun09g079900* encoding the MYB transcription factors was noted as being putative.

## QTL comparison

Out of the 30 QTLs identified for seed eye color, seed coat color patterns, and seed coat texture, 14 were novel. The novel QTL included *qNo_eye-5–1, qblk_eye-7–1, qbr-eye-7–1* for seed eye color,*qblk_S-7–1, qblk_S-9–1, qbr_S-7–1, qbr_S-9–1, qEy2-7-1, qEy2-8-1 qHol-7–1, qfull_C-9–1,* for seed coat color and patterns; and three additional QTLs located on chromosomes 8 and 10 for seed coat texture (Table 4). The rest of the QTLs were positioned closely on the same chromosomes to previously reported QTLs.

## Relative efficiency of marker-assisted selection (RE-MAS)

The relative efficiency (RE) of marker-assisted selection (MAS) and marker-based selection (MBS) with major QTLs effects compared with phenotypic selection (PS) were estimated (Table 5). Phenotypic accuracy was consistently high (0.79–0.80) across traits, with observed heritability values ranging from 0.622 (Holstein) to 0.633 (Smth and Rgth). The RE of MAS compared to PS (RE-MAS vs PS) was stable across traits (1.26–1.27), suggesting an interest of combining marker and phenotypic information over phenotypic selection alone. By contrast, the RE of MBS relative to PS (RE_MBS vs PS) ranged more widely (0.59–1.26), explaining that marker-only selection is less efficient for some traits (br_eye and br_S), but can approach or exceed phenotypic selection for others (Holstein, and full_coat).

## Discussion

Seed coat is the external protective layer, created from the integuments of the ovule post-fertilization. Its role is to safeguard the seed from physical harm, dehydration, and often from insects and pathogens. It plays a crucial role in the seed's viability and germination process [31]. Seed coat appearance is a key differentiating parameter of the cowpea market segments, and it constitutes one of the major factors that consumers consider in making choices of which grain type to buy in the market. Breeders are therefore careful in their selection process to ensure that the seed coats of their improved varieties possess the market desired appearance. Seed coat texture in cowpea is very important especially when it comes to how the grain is to be consumed or processed. For the boiled grain MS, it is important that the seed coat remains intact after cooking. Rough seed coat is also preferred in this MS because it allows fast cooking possibly due to easy water imbibition [32]. In the home-made flour MS, testa dehulling is important and rough seed coats facilitate easier testa removal when the grain is to be milled into flour or dough. However, for the industrial flour MS, the focus is on high fiber content of the cowpea testa and the benefit of fiber in human nutrition [33], hence, seed coat texture is not important since the entire grain is ground into flour. In the Industrial flour MS, eye color and seed coat color are important as these can affect the appearance of the flour. Consequently, eyeless and white seed coats are preferred for the industrial flour MS.

Pigmentation can be found across various cowpea plant parts, such as, nodal joints, stems, leaves, peduncles, flower petals, cotyledons, seeds, and pods, [34]. The coloration of the plant parts is due to anthocyanins, which are pigments that play key roles in plants' protection. The pigments are also responsible for seed coat coloration, with different molecules contributing to the diversity of colors and influencing the commercial value of cowpea beans [35,36].

A genetic linkage map showing markers tightly linked to desired traits is essential for molecular breeding. QTL analysis serves as an effective practical approach to this and helps identify the genomic regions associated with traits of interest in crops [37].

Previous studies on the inheritance of cowpea seed morphological characters have reported that two pairs of independent recessive genes confer rough seed coat texture, while the presence of at least one dominant gene at each of

**Table 4. Summary of QTL comparison.**

| Trait | QTL in present Study | | | Previously reported QTL | | | Ref |
|---|---|---|---|---|---|---|---|
| | QTL | Chr | Pos (bp) | QTL | Chr | Pos (bp) | |
| Eyeless | qNo_eye-5–1 | 5 | 42,906,632 | – | – | | Novel |
| | qNo_eye-7–1 | 7 | 22,202,114 | Color Factor | Vu07 | 20,544,306 | [8] |
| | qNo_eye-7–2 | 7 | 7105402 | Color Factor | Vu07 | _ | [8] |
| Eye 2 | qEy2-7-1 | 7 | 21,498,532 | – | – | – | Novel |
| | qEy2-8-1 | 8 | 31,380,509 | – | – | – | Novel |
| | qEy2-10-1 | 10 | 38,417,291 | Eye 2 | Vu10 | 36,854,255 | [8] |
| | qEy2-10-2 | 10 | 34,686,751 | Eye 2 | Vu10 | 36,854,255 | [8] |
| Holstein | qHol-7–1 | 7 | 21,498,532 | – | – | – | Novel |
| | qHol-9–1 | 9 | 30,312,420 | Holstein | Vu09 | 29,253,230 | [29] |
| | qHol-10–1 | 10 | 38,417,291 | Holstein | Vu10 | SNP2_2435 and;39,484,624 | [8,29] |
| Watson | qWat-7–1 | 7 | 21,498,532 | Watson | vu07 | _ | [30] |
| | qWat-10–1 | 10 | 38,354,471 | Watson | Vu10 | 36,854,255 | [8] |
| Full coat | qfull_C-7–1 | 7 | 21,498,532 | Full coat | vu07 | _ | [8] |
| | qfull_C-9–1 | 9 | 31,055,562 | _ | _ | _ | Novel |
| | qfull_C-10–1 | 10 | 38,417,291 | Full coat | Vu10 | 37,148,627 | [8] |
| Seed black eye | qblk_eye-5–1 | 5 | 2.665,433 | black eye | vu05 | 3,015,637, | [29] |
| | qblk_eye-7–1 | 7 | 21,498,532 | | | | novel |
| Seed brown eye | qbr-eye-7–1 | 7 | 21,498,532 | _ | _ | | novel |
| Black seed | qblk_S-5–1 | 5 | 2,665,433 | Black seed | vu05 | _ | [27] |
| | qblk_S-7–1 | 7 | 21,498,532 | _ | _ | | novel |
| | qblk_S-9–1 | 9 | 9,329,245 | _ | _ | | novel |
| Brown seed | qbr_S-7–1 | 7 | 21,498,532 | _ | _ | | novel |
| | qbr_S-9–1 | 9 | 30,800,118 | _ | _ | | novel |
| White seed | qwht_S-7–1 | 7 | 21,498,532 | White seed | vu07 | 21,041,365 | [29] |
| Seed coat texture | qSmh-8–1 | 8 | 34,800,203 | _ | _ | _ | Novel |
| | qSmh-8–2 | 8 | 204,958 | _ | _ | _ | Novel |
| | qSmh-10–1 | 10 | 38,701,443 | _ | _ | _ | Novel |
| | qRgh-8–1 | 8 | 34,800,203 | _ | _ | _ | Novel |
| | qRgh-8–2 | 8 | 204,958 | _ | _ | _ | Novel |
| | qRgh-10–1 | 10 | 38,701,443 | _ | _ | _ | Novel |

the two loci results in smooth seed coat [9,38]. In our study, however, we identified three major QTLs on Chr8 and Chr10 associated with seed coat texture. This represents the first mapping of QTLs associated with seed coat texture in cowpea, marking a significant advancement in understanding this trait's genetic regulation. Comparable studies carried out in other grain legumes identified five QTLs in soybean and twenty-seven QTLs in black common bean for seed coat texture [39].

The expressions of different seed coat colors, and seed coat color patterns in the RILs which are beyond the ranges found in the two parental lines clearly indicate transgressive segregations for these traits. This finding demonstrates that seed coat color patterns in cowpea is polygenic, with genes often exhibiting epistatic actions. The research on cowpea seed coat color and distribution patterns is both challenging and interesting because of its complex genetic basis and broad implications for food science and agriculture. This complexity stems from the development of seed coat, in which the ovules' integuments differentiate into specialized cell types through a complicated network of interactions between genotype and environment [21,40,41]. Unlike seed coat texture, multiple genes are involved in controlling seed coat color

**Table 5. Genetic variance, heritability, marker variance, and relative efficiency of marker-assisted selection (RE-MAS) for major QTLs detected. VA: additive variance, h²: heritability, MAS: marker-assisted selection, PS: phenotypic selection, MBS = Marker-base selection.**

| Trait | QTL | Trait-prevalence | VA_ | h2_ | VM_ | Pheno_accuracy | RE_MAS_vs_PS | RE_MBS_vs_PS |
|---|---|---|---|---|---|---|---|---|
| No_eye | qNo_eye-5–1 | 0.32 | 0.158 | 0.630 | 0.12 | 0.79 | 1.26 | 1.12 |
| Ey2 | qEy2-7-1 | 0.17 | 0.157 | 0.629 | 0.09 | 0.79 | 1.26 | 0.96 |
| Holstein | qHol-7–1 | 0.16 | 0.156 | 0.622 | 0.15 | 0.79 | 1.27 | 1.26 |
| Watson | qWat-7–1 | 0.17 | 0.157 | 0.630 | 0.10 | 0.79 | 1.26 | 0.99 |
| full_coat | qfull_C-7–1 | 0.19 | 0.157 | 0.628 | 0.12 | 0.79 | 1.26 | 1.11 |
| blk_eye | qblk_eye-5–1 | 0.37 | 0.158 | 0.632 | 0.11 | 0.80 | 1.26 | 1.06 |
| br_eye | qbr-eye-7–1 | 0.31 | 0.158 | 0.631 | 0.03 | 0.79 | 1.26 | 0.59 |
| blk_s | qblk_S-5–1 | 0.28 | 0.157 | 0.627 | 0.10 | 0.79 | 1.26 | 0.99 |
| br_S | qbr_S-7–1 | 0.24 | 0.157 | 0.629 | 0.04 | 0.79 | 1.26 | 0.60 |
| Wht_S | qwht_S-7–1 | 0.48 | 0.158 | 0.631 | 0.05 | 0.79 | 1.26 | 0.72 |
| Smth | qSmh-8–1 | 0.59 | 0.158 | 0.633 | 0.08 | 0.80 | 1.26 | 0.88 |
| Rgth | qRgh-8–1 | 0.41 | 0.158 | 0.633 | 0.08 | 0.80 | 1.26 | 0.90 |

and patterns of distribution, as reported in previous studies focused on mapping QTLs with effects on these cowpea traits. Our findings align with previous studies that seed coat pigmentation in cowpea is regulated by a multi-locus system [42–44]. Earlier studies have identified three primary gene systems, C (Color Factor), W (Watson), and H (Holstein) with simple dominance and epistatic interactions to produce various seed coat colors and patterns in cowpea [8,13,45] The C gene encodes for a "constriction" factor, while the W and H loci genes code for different "expansion" factors. The C locus represents the main locus determining the seed coat pattern.

The H and W loci affect the extent of the distribution, and their impact is only apparent with an unconstrained allele (C2) at the C locus. The Holstein and Watson patterns manifest when Holstein (H1) and Watson (W0) are present. Watson (W1) and Holstein (H1) traits work together to produce the Full Coat phenotype. These loci collaborate to determine the various patterns and colors observed in cowpea seed coat, including the distribution and intensity of pigmentation. Accordingly, we employed the Herniter et al. [8] and Spillman [14] classification methods of seed coat color and pattern types to categorize the observed main seed coat types in the RIL population. Based on the above proposed allelic series, an individual with the $C_0C_0$ genotype will express no color (no eye) pattern, regardless of the W and H loci genotypes. The possible genotypes for no color pattern are: $C_0C_0W_0W_0H_0H_0$, $C_0C_0W_1W_1H_0H_0$, $C_0C_0W_0W_0H_1H_1$, and $C_0C_0W_1W_1H_1H_1$. However, in the absence of constricted $C_0$ alleles, the expansion factor can be observed. An individual with $C_2C_2W_0W_0H_0H_0$ will express the Eye 2 pattern. An individual with $C_2C_2W_0W_0H_1H_1$ genotype will express the Holstein pattern, while an individual with the $C_2C_2W_1W_1H_0H_0$ genotype will express the Watson pattern. An individual with the $C_2C_2W_1W_1H_1H_1$ genotype expresses a Full Coat pattern [13]. White seed coat represented the highest proportion, and three eye color types were observed: no color, black color, and brown color. QTLs linked to black eye color were identified on Chr5 and Chr7, aligning partly with previous findings on Chr5 [29]. A major QTL for brown eye color on Chr7 differed from previous studies that found the QTL to be located on Chr3 and Chr10 [29]. QTLs for black seed coat were identified on Chr5, Chr7, and Chr9, while only one had previously been identified on Chr5 [27], Two QTLs on Chr7 and Chr9 were identified as being associated with brown seed color, while Xiong et al. [29] reported the presence of two loci on Chr5 and Chr8 to be responsible for the expression of brown seed coat color. The single locus identified as responsible for white seed on Chr7 is probably the same as reported by Xiong et al. [29]. Notably, a major QTL for the no pigmentation of the seed coat trait in the region with clustered QTLs on Chr7 aligns with results from previous studies [8,29], while the QTLs on Chr5 seem to be novel. Multiple traits were clustered in overlapping regions, especially on Chr7 and Chr10, indicating possible pleiotropic effects. Most of the previous studies have been focused on

the identification of QTLs associated with important traits in cowpea, using single-locus analyses and linkage mapping [19,20,46,47]. However, fewer studies [48] have explored the role of epistatic interactions in shaping important traits of cowpea, despite evidence from other legumes suggesting that epistasis plays a crucial role in their trait expression and heritability [49]. However, epistatic interaction effects have been shown to be key genetic factors that contribute significantly to the phenotypic variation observed in complex traits. Understanding these interactions is essential for understanding the genetic mechanisms underlying complex traits [50–52]. In addition to the main-effect QTLs, our study identified several significant digenic epistatic interactions associated with seed coat pigmentation, suggesting that this trait is governed by complex genetic relationships beyond additive effects. The epistatic QTLs are valuable for designing marker-assisted selection (MAS) strategies, especially when pyramiding favorable alleles [49]. We report 116 pairs of digenic epistatic QTLs, explaining up to 62.04% of the phenotypic variation influencing seed coat color and pattern, highlighting the importance of gene interactions in the regulation of these traits. Several traits showed significant epistatic interactions in Chr5, Chr7, Chr9, and Chr10, demonstrating complex genetic interactions where the phenotypic expression of seed coat pigmentation is controlled by multiple genes. These non-additive genetic interactions, particularly concentrated on certain regions, indicate coordinated regulatory networks controlling multiple seed traits, consistent with both epistatic and pleiotropic effects. As mentioned earlier, the seed coat pigmentation is due to flavonoid biosynthesis regulation, and this has been studied in numerous crops [17,35]. The flavonoid biosynthesis is regulated by a transcription factor family composed of a MYB protein, a WD-repeat protein, and a basic helix-loop-helix (bHLH) protein [28]. Whereas the E3 ubiquitin domain was reported as a negative regulator of anthocyanin biosynthesis [53]. In the present study, twelve putative genes associated with eye color, seed coat color, seed coat color patterns were located on Chr5, Chr7, Chr9, and Chr10. These genes coded for MYB transcription factors, WD40-repeat-containing domain, Myc-type, basic helix-loop-helix (bHLH) domain, AP2/ERF domain-containing protein, WRKY domain, and Ubiquitin-related domain. The identification of these putative genes was consistent with the results of some previous studies. For instance, Herniter et al. [8] reported the helix-loop-helix protein as controlling the No eye Color, Eye 1, and Full Coat pattern on Chr7, the WD-repeat gene controlling the Eye 2, Holstein, Watson, and Full Coat on Chr9, and an E3 ubiquitin ligase controlling the Eye 1, Eye 2, Holstein, Watson, and Full Coat on Chr10. The MYB Transcription gene regulating flavonoid biosynthesis was reported by Herniter et al. [27] as controlling purple pod tip and black seed coat colors. Myc-type, basic helix-loop-helix (bHLH) domain was noted as a candidate gene regulating flower color in cowpea, suggesting a possible dual role of the gene [46]. The genes, including ERF domain-containing protein and the WRKY domain, were reported to control the flavonoid biosynthesis pathway in some crops [28].

The estimation of the relative efficiency of marker-assisted selection (RE-MAS) provided valuable insights into the potential use of genomic tools for improving seed coat appearance in cowpea. The results of the relative efficiency (RE) analysis showed differences among traits regarding the potential advantage of marker-assisted selection (MAS) over phenotypic selection alone. Interestingly, for all traits, RE values greater than 1 were observed, indicating that MAS is predicted to be more efficient than phenotypic selection. This trend was particularly evident for all the traits suggesting that genetic markers linked to these traits explain a substantial proportion of the genetic variance and can enhance selection accuracy. The observation that MAS outperformed phenotypic selection for several traits demonstrates its potential to accelerate selection, reduce breeding cycle time, and enhance accuracy, especially for traits that are difficult to score or highly influenced by the environment [54]. In summary, these findings provide valuable insights into the molecular mechanism underlying seed coat appearance of cowpea. The identified genetic loci and putative genes give a foundation for enhancing the measured traits, which can support future breeding efforts aimed at improving cowpea for consumer and market demands. Major QTLs with PVE ≥ 10% represent promising targets for MAS. The co-localization of pigmentation trait QTLs on chromosomes 7 and 10 may enhance breeding efficiency by enabling selection for multiple preferred traits using a single marker set. However, further molecular investigation will be essential to validate these findings for their possible use in marker assisted selection (MAS) in future cowpea breeding.

## Supporting information

**S1 Fig. QTLs positions of seed coat traits (A) Seed coat color pattern (B) Seed coat texture.**
(DOCX)

**S1 Table. Phenotypic data collected.**
(XLSX)

**S2 Table. SNP-marker distribution on cowpea linkage map.**
(XLSX)

**S3 Table. QTLs identified for seed coat appearance per year.**
(XLSX)

**S4 Table. List of epistatic QTLs for coat color pattern.**
(XLSX)

**S5 Table. List of possible candidate genes regulating seed coat pigmentation.**
(XLSX)

## Acknowledgments

The authors appreciate all the staff of Cowpea Breeding Unit of the International Institute of Tropical Agriculture, Ibadan, Nigeria, for the technical support provided in establishing the field trials.

## Author contributions

**Conceptualization:** Abdoul Moumouni Iro Sodo, Christian Fatokun, Ousmane Boukar.

**Data curation:** Abdoul Moumouni Iro Sodo, Patrick Obia Ongom.

**Formal analysis:** Abdoul Moumouni Iro Sodo, Patrick Obia Ongom, Ibnou Dieng.

**Funding acquisition:** Abdoul Moumouni Iro Sodo, Patrick Obia Ongom, Ousmane Boukar.

**Investigation:** Abdoul Moumouni Iro Sodo, Ousmane Boukar.

**Methodology:** Abdoul Moumouni Iro Sodo, Christian Fatokun, Bunmi Olasanmi, Ibnou Dieng, Ousmane Boukar.

**Resources:** Ousmane Boukar.

**Software:** Abdoul Moumouni Iro Sodo.

**Supervision:** Bunmi Olasanmi, Ousmane Boukar.

**Visualization:** Abdoul Moumouni Iro Sodo.

**Writing – original draft:** Abdoul Moumouni Iro Sodo.

**Writing – review & editing:** Abdoul Moumouni Iro Sodo, Bunmi Olasanmi, Patrick Obia Ongom, Ibnou Dieng, Ousmane Boukar.

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
