## [Decision Letter · Decision Letter 0]

2 Jul 2025

PLOS ONE

Dear Dr. Moumouni,

Thank you for submitting your manuscript to PLOS ONE. After careful consideration, we feel that it has merit but does not fully meet PLOS ONE’s publication criteria as it currently stands. Therefore, we invite you to submit a revised version of the manuscript that addresses the points raised during the review process.

We look forward to receiving your revised manuscript.

Kind regards,

Karthikeyan Adhimoolam

Academic Editor

PLOS ONE

 [AVISA project, i.e. Accelerated varietal improvement and seed delivery of legumes and dryland cereals in Africa (AVISA) - AVISA-Transition project. The investment ID is INV-049752]. 

3. In the online submission form, you indicated that [The original contributions presented in the study are included in the paper/Supplementary Material. For any additional questions, please reach out to the corresponding author.].

Additional Editor Comments (if provided):

Reviewers' comments:

Reviewer's Responses to Questions

**Comments to the Author**

1. Is the manuscript technically sound, and do the data support the conclusions?

Reviewer #1: Yes

Reviewer #2: Yes

2. Has the statistical analysis been performed appropriately and rigorously?

Reviewer #1: Yes

Reviewer #2: Yes

3. Have the authors made all data underlying the findings in their manuscript fully available?

Reviewer #1: Yes

Reviewer #2: Yes

4. Is the manuscript presented in an intelligible fashion and written in standard English?

Reviewer #1: Yes

Reviewer #2: Yes

Reviewer #1: Text attached in attachement.

Weaknesses and Suggestions for Improvement

a) Clarify Seed Coat Trait Definitions

• Provide more detail on how seed coat pigmentation and texture were scored. Include whether digital imaging, colorimetric scores, or visual ratings were used.

b) Statistical Thresholds and Methods

• Clearly state the statistical models, QTL significance thresholds (e.g., LOD cutoff), and software used (e.g., R/qtl, QTL IciMapping, TASSEL). If permutation tests were used, please report how many iterations were performed.

c) Effect Sizes of QTLs

• Although QTLs with PVE (phenotypic variance explained) are reported, the manuscript should emphasize which QTLs are stable and robust across years/environments.

d) Candidate Gene Annotations

• A table summarizing the 12 candidate genes with gene IDs, putative functions, chromosomal location, and references will strengthen the biological relevance. Consider using functional enrichment analysis or expression data if available.

e) Epistasis Interpretation

• Provide a figure or matrix summarizing epistatic QTL pairs and highlight any biologically meaningful interactions. Discuss the potential breeding implications of these interactions, even if minor.

f) Language and Technical Corrections

• Improve grammatical clarity throughout. For example:

o "mapbased cloning" → "map-based cloning"

o "quality DArTag single nucleotide polymorphisms (SNPs)" → consider simplifying to "a panel of 2602 high-quality DArTag SNPs"

4. Minor Suggestions

• Include a figure showing representative seed coat patterns of parental lines (RP270 and CB27) and selected RILs.

• Ensure consistent usage of units, e.g., percentages for PVE.

• All abbreviations (e.g., QTL, SNP, RIL) should be defined at first use in the manuscript.

Reviewer #2: Manuscript by Moumouni et al describes identification and analysis of QTLs with effects on seed coat appearance in cowpea. The work is interesting; authors identified 30 major QTLs associated with seed coat color and texture. Some candidate genes involved in the control of these traits for grain color traits were identified.

In general, this paper is well written, most methods are described properly. Results description is sufficient. However, I have some comments that should be clarified.

Line 160: “reference genome (IT97K-499-35)”. Please provide literature reference for genome assembly.

Lines 160-162: “Based on gene ontology and available literature, the possible putative genes were selected based on their participation in the flavonoid biosynthesis pathway.”

The description of methods for gene prioritization is not sufficient. Please provide the following information:

What is the source of the Gene ontology terms for genes?

What literature sources were used to make prioritization?

Why flavonoid biosynthesis pathway only was used for prioritization?

What are criteria for gene “participation in the flavonoid biosynthesis pathway”?

I also recommend to move some partial description of prioritization procedure from lines 257-259.

There is some description of the flavonoid biosynthesis pathway genes in lines 370-376 (Discussion section). I recommend to move it to methods section also (to group all the description of prioritization methods in one place). I noticed that genes from this pathway represented by regulatory proteins only. What about enzymes participating in the flavonoid biosynthesis?

Line 257: “Phytozome and cowpea database”. The version of the Phytozome database and literature reference should be provided. What “cowpea database” was used for data analysis?

Line 259: “considering the level of relative expression of the flavonoid biosynthesis pathway.” This phrase is not clear. What expression data were used? How authors determined genes of the flavonoid biosynthesis pathway? How their relative expression was determined? This should be clearly described in the methods section.

Some figures have low quality.

Fig 2: Text in the figure is almost unreadable. I don’t see chromosome names in the figure. Panels have no letters and are not aligned properly.

Fig 3: The information in the ovals on the chromosomes and numbers (?) close to the chords is unreadable.

Comments on additional materials:

Figure “S1 Fig.QTLs positions of seed coat traits (A) Seed coat colour and pattern (B) Seed coat texture.tif”. The drawing is of poor quality, and the text cannot be read. I recommend increasing the resolution, or, if possible, using a vector file format (SVG or PDF).

File “S2 Table. Epistatic interactions of QTLs linked to cowpea seed eye color, seed coat color and seed coat color patterns.xlsx” does not open properly: it is reported that related files are missing.

File “S3 Table. Retrieved genes from Phytozome.xlsx”: I recommend adding a column indicating the chromosome on which the markers and genes are located.

**Do you want your identity to be public for this peer review?** For information about this choice, including consent withdrawal, please see our Privacy Policy

Reviewer #1: **Yes: ** Dr. Muraleedhar S. Aski

Reviewer #2: No

---

## [Author Response · Author response to Decision Letter 1]

15 Jul 2025

Dear Editors-in-Chief,

I am writing to submit the revised version of the manuscript titled "Identification of QTLs with effects on seed coat appearance in cowpea" following reviewers’ comments and suggestions received from PLOS ONE Journal

We really appreciate the editor and the reviewers’ positive and insightful comments and suggestions, which have enabled us to improve the write-up. Below, we provide a point-by-point response to each reviewer’s comment. All changes have been incorporated into the revised version of the manuscript, with track changes for clarity.

i. Editor requests

- The role of the funder: The funder of the study reported in this manuscript, the Accelerated Varietal Improvement and Seed Delivery of Legumes and Dryland Cereals in Africa (AVISA), had no role in the study design, data collection and analysis, decision to publish, or preparation of the manuscript.

i. Editor’s requests

- The role of the funder: "The funder s of the study reported in this manuscript, the Accelerated Varietal Improvement and Seed Delivery of Legumes and Dryland Cereals in Africa (AVISA), had no role in the study design, data collection and analysis, decision to publish, or preparation of the manuscript.

- Laboratory protocols

We appreciate the editor's comments that we should submit the laboratory protocols to support reproducibility. However, in our case, this recommendation is not applicable. We did not perform laboratory-based DNA extraction or genotyping ourselves; the leaf samples were collected and prepared in accordance with the standard procedures required by the genotyping company (Intertek Agritech laboratory) as described in the Materials and Methods section. We have ensured that all steps under our control, especially the sampling procedures, data collection, and data analyses, are thoroughly described in the manuscript to allow reproducibility.

- Data availability

We really acknowledge PLOS ONE’s policy on open data sharing to allow transparency and reproducibility. The phenotypic data and all QTL results generated in the present study have been made available in the manuscript and supporting information. The genotypic data used are currently part of a larger ongoing research project and can be accessed by contacting the principal investigator at O.Boukar@cgiar.org and eventually will become available through the IITA open access database at https://data.iita.org/.

ii. Reviewer 1’s comments

a) Clarify Seed Coat Trait Definitions

We have revised the Materials and Methods section (highlighted area in lines 127-129) to provide a clearer description of the scoring of the seed coat pigmentation and seed coat texture.

b) Statistical Thresholds and Methods

We have revised the Materials and Methods section (in lines 158-161) to provide detailed information on the QTL mapping. QTL analysis was conducted using QTL IciMapping version 4.2. For both main-effect and epistatic QTL detection, we performed permutation tests with 1000 iterations to determine appropriate LOD thresholds for significance, as estimated by the software.

c) Effect Sizes of QTLs

We really appreciate the reviewer’s suggestion on stable QTL. However, we did not carry out separate QTL analyses for each year because the phenotypic data collected across the two seasons were highly consistent, reflecting the genetic stability of the RIL population used. As a result, we performed a QTL analysis using the combined data from both years that enabled us to increase the statistical power and detect QTLs that represent the overall genetic effects across years. However, in response to the reviewer’s comment, we have now conducted separate QTL analyses for each year (S3 Table). The results confirm that the major QTLs identified in the combined analysis are also consistently detected in both individual years.

d) Candidate Gene Annotations

We have added a new table (Table 3) in Lines-287-289 summarizing the candidate genes, including gene IDs, chromosomal number, predicted functions, and supporting references.

e) Epistasis Interpretation

We appreciate the reviewer’s insightful comment on the importance of visualizing and interpreting epistatic interactions. In the original submission (Figure 4) is provided, that shows the digenic epistatic QTL interactions. This figure presents the significant epistatic QTL pairs detected across the 11 chromosomes of cowpea and highlights potential functional interactions relevant to seed coat pigmentation. To further strengthen the biological relevance of these findings, we have now provided a minor discussion on the potential breeding implications of these epistatic interactions in the revised manuscript (Discussion section, from lines 376 to 380).

Minor Suggestions

Regarding the inclusion of a figure showing representative seed coat patterns of parental lines (RP270 and CB27) and selected RILs, we would like to kindly note that this figure was already included in the initial submission as Figure 1. It presents the seed coat patterns of the two parental lines (RP270 and CB27) alongside selected RILs.

We have checked the manuscript and ensured that all abbreviations are clearly defined upon their first mention.

Reviewer 2’s comments

We thank the reviewer for his suggestions, and the following details have been added from lines 261 to 263:

- We added the appropriate reference for the Vigna unguiculata (IT97K-499-35) in lines 167-168.

- The genes were retrieved from the Phytozome v1.1 database (http://phytozome.jgi.doe.gov/), and the InterPro portal was used for gene models along with their functional annotations.

- For the flavonoid biosynthesis pathway, no original expression data were generated in this study; we only used expression pattern references where available.

- We have revised Supplementary Tables S2 and S3 and increased the figure resolution.

- The names of the QTLs, chromosome numbers, and connecting lines in Figure 3 are fully visible when the PDF version of the manuscript is zoomed in, as the figures were submitted in high resolution.

The manuscript has been thoroughly revised and agreed upon by all authors.

Thank you for your time and consideration. We look forward to hearing from you soon.

Sincerely,

Abdoul Moumouni Iro Sodo

International Institute of Tropical Agriculture

abdoulmoumouniirosodo@gmail.com

---

## [Decision Letter · Decision Letter 1]

12 Aug 2025

Dear Dr. Moumouni,

Thank you for submitting your manuscript to PLOS ONE. After careful consideration, we feel that it has merit but does not fully meet PLOS ONE’s publication criteria as it currently stands. Therefore, we invite you to submit a revised version of the manuscript that addresses the points raised during the review process.

We look forward to receiving your revised manuscript.

Kind regards,

Karthikeyan Adhimoolam

Academic Editor

PLOS ONE

Journal Requirements:

Additional Editor Comments:

** ** I suggest the authors carefully follow the reviewers comments and make the necessary corrections to the manuscript. 

Reviewers' comments:

Reviewer's Responses to Questions

**Comments to the Author**

Reviewer #1: All comments have been addressed

Reviewer #2: (No Response)

2. Is the manuscript technically sound, and do the data support the conclusions?

Reviewer #1: Yes

Reviewer #2: Yes

3. Has the statistical analysis been performed appropriately and rigorously?

Reviewer #1: Yes

Reviewer #2: Yes

4. Have the authors made all data underlying the findings in their manuscript fully available?

Reviewer #1: Yes

Reviewer #2: Yes

5. Is the manuscript presented in an intelligible fashion and written in standard English?

Reviewer #1: Yes

Reviewer #2: Yes

Reviewer #1: Thank you for the revisions. Few points which will add to the value of the manuscript.

Length and density of abstract.

Slightly word-heavy for an abstract; some sentences could be shortened for more impact.

High number of numeric details in one paragraph can overwhelm the reader.

Lack of novelty emphasis

Does not explicitly highlight what is new compared to previous studies on cowpea pigmentation QTLs.

One major question: "What is the advancement beyond earlier mapping studies?" plese mention.

Limited statistical clarity

The term “major QTLs” is used, but the LOD thresholds, confidence intervals, or effect size definitions are not given (though some of this may be fine to omit for space).

“Much smaller effect” for epistatic QTLs is qualitative; could be backed with a range or average.

Trait definition could be sharper

“Seed coat appearance traits” is broad—could briefly define if this includes coat color, eye pattern, mottling, etc.

Currently, pigmentation focus is clear, but texture (mentioned in the intro) is not addressed in the results.

Candidate gene evidence

Mentions 12 candidate genes from GO/literature, but does not say whether any are novel or validated.

Lacks mention of functional categories (e.g., flavonoid biosynthesis, anthocyanin pathway) that could make it more engaging.

No quantitative marker-assisted selection potential

While breeding application is stated, the potential for predictive accuracy or MAS efficiency is not quantified.

Reviewer #2: Some questions were not clarified by authors:

What literature sources were used to make prioritization?

Why flavonoid biosynthesis pathway only was used for prioritization?

What are criteria for gene “participation in the flavonoid biosynthesis pathway”?

I recommend to clarify these questions in the methods section (QTL analysis and candidate gene identifications)

Authors replied: "- For the flavonoid biosynthesis pathway, no original expression data were generated in this

study; we only used expression pattern references where available."

Please provide the source of the gene expression information (database or literature, etc)

**Do you want your identity to be public for this peer review?** For information about this choice, including consent withdrawal, please see our Privacy Policy

Reviewer #1: No

Reviewer #2: No

---

## [Author Response · Author response to Decision Letter 2]

27 Aug 2025

Dear Editors-in-Chief,

I am writing to submit the revised version of the manuscript titled "Identification of QTLs with effects on seed coat appearance in cowpea" following reviewers’ comments and suggestions received from PLOS ONE Journal

We really appreciate the editor and the reviewers’ insightful comments and suggestions, which added value of the manuscript. Below, we provide a point-by-point response to each reviewer’s comment. All changes have been incorporated into the revised version of the manuscript, with track changes for clarity.

i. Editor requests

We are grateful for insightful comments. We have crosschecked and reviewed the reference list and confirmed that it is complete, correctly formatted, and that none of the cited reports has been retracted and no additional citation was requested by the reviewers.

Reviewers’ comments

Length and density of abstract.

Comment: Slightly word-heavy for an abstract; some sentences could be shortened for more impact. High number of numeric details in one paragraph can overwhelm the reader.

Response: We appreciate the observation and have condensed the abstract by merging related sentences (Line14-line16) and (line 20-line21).

Lack of novelty emphasis

Comment: Does not explicitly highlight what is new compared to previous studies on cowpea pigmentation QTLs. One major question: "What is the advancement beyond earlier mapping studies?" please mention.

Response: We appreciate the comment. We have emphasized that our study reports the first mapping of QTLs for seed coat texture in cowpea in the introduction indicating (line 94-line 96). For Identification of novel, to address this, we have now added a new Table 4 (line 309-line 317) comparing the QTLs identified in this study with those reported previously in cowpea seed coat appearance studies. These comparative highlights that 14 out of the 30 QTLs detected are novel, providing strong evidence of advancement beyond earlier mapping efforts and the integration of epistatic QTL analysis, revealing 116 significant interactions, which has rarely been done for these traits in cowpea (Line 266-line 269).

Limited statistical clarity

Comment: The term “major QTLs” is used, but the LOD thresholds, confidence intervals, or effect size definitions are not given (though some of this may be fine to omit for space). “Much smaller effect” for epistatic QTLs is qualitative; could be backed with a range or average.

Response: We appreciate the comment, and we now define major QTLs as those explaining ≥10% phenotypic variance (PVE) in methodology (line 162-line 164) as per standard QTL mapping threshold. In the Results section, we provide average and range of PVE for epistatic QTLs (line 268-line 270).

Trait definition could be sharper

Comment: “Seed coat appearance traits” is broad—could briefly define if this includes coat color, eye pattern, mottling, etc.

Response: We agree and have defined early in the Methodology (line123-126) that seed coat appearance traits in present study include seed coat color, seed eye color, seed coat color pattern, and seed coat texture.

Comment: Currently, pigmentation focus is clear, but texture (mentioned in the intro) is not addressed in the results.

Response: Thank you for the observation; the seed texture has been addressed; the QTL for texture is presented in Table 2 and the texts in lines 248-249 specifies the number of QTLs identified for seed coat texture.

Candidate gene evidence

Comment: Mentions 12 candidate genes from GO/literature but does not say whether any are novel or validated. Lacks mention of functional categories (e.g., flavonoid biosynthesis, anthocyanin pathway) that could make it more engaging.

Response: We have grouped the 12 candidate genes into three functional categories flavonoid biosynthesis, transcriptional regulation, and ubiquitin-mediated regulation, and indicated which genes are novel in cowpea pigmentation studies (lines-440-444).

Marker-assisted selection (MAS) potential

Comment: While breeding application is stated, the potential for predictive accuracy or MAS efficiency is not quantified.

Response: Thank you for this insightful observation. In response, we have now conducted a Relative Efficiency of Marker-Assisted Selection (RE-MAS) analysis for the major QTLs identified in this study showing the comparative efficiency of MAS relative to phenotypic selection (Line 170-Lines 189). These results provide a quantitative assessment of the breeding value of the identified QTLs and strengthen the practical application of our findings for marker-assisted selection in cowpea improvement Table 5 in lines 318-331.

The discussion section has also been updated to highlight the implications of these results for breeding programs (lines 445-455).

The manuscript has been thoroughly revised and agreed upon by all authors.

Thank you for your time and consideration. We look forward to hearing from you soon.

Sincerely,

Abdoul Moumouni Iro Sodo

International Institute of Tropical Agriculture

abdoulmoumouniirosodo@gmail.com

---

## [Decision Letter · Decision Letter 2]

14 Sep 2025

Identification of QTLs with effects on seed coat appearance in cowpea

PONE-D-25-27297R2

Dear Dr. Moumouni,

We’re pleased to inform you that your manuscript has been judged scientifically suitable for publication and will be formally accepted for publication once it meets all outstanding technical requirements.

Kind regards,

Karthikeyan Adhimoolam

Academic Editor

PLOS ONE

Additional Editor Comments (optional):

Reviewers' comments:

Reviewer's Responses to Questions

**Comments to the Author**

Reviewer #1: All comments have been addressed

Reviewer #2: All comments have been addressed

2. Is the manuscript technically sound, and do the data support the conclusions?

Reviewer #1: Yes

Reviewer #2: Yes

3. Has the statistical analysis been performed appropriately and rigorously?

Reviewer #1: Yes

Reviewer #2: Yes

4. Have the authors made all data underlying the findings in their manuscript fully available?

Reviewer #1: Yes

Reviewer #2: Yes

5. Is the manuscript presented in an intelligible fashion and written in standard English?

Reviewer #1: Yes

Reviewer #2: Yes

Reviewer #1: I am satified with the revision, I would like to formally accept the decision regarding my revised manuscript.

Reviewer #2: The paper can be accepted for publication, authors addressed comments from reviewers. The manuscript was substantially improved.

**Do you want your identity to be public for this peer review?** For information about this choice, including consent withdrawal, please see our Privacy Policy

Reviewer #1: **Yes: ** Muraleedhar Aski

Reviewer #2: No

---

## [Editor Report · Acceptance letter]

PONE-D-25-27297R2

PLOS ONE

Dear Dr. Moumouni,

I'm pleased to inform you that your manuscript has been deemed suitable for publication in PLOS ONE. Congratulations! Your manuscript is now being handed over to our production team.

Kind regards,

on behalf of

Dr. Karthikeyan Adhimoolam

Academic Editor

PLOS ONE